# SeasonBench-EA: A Multi-Source Benchmark for Seasonal Prediction and Numerical Model Post-Processing in East Asia

**Mengxuan Chen**
Tsinghua University

**Guowen Li**
Sun Yat-sen University

**Ziheng Zou**
Tsinghua SIGS

**Fang Wang**
CMA Earth System Modeling and Prediction Centre (CEMC)

**Jinxiao Zhang**
Tsinghua University

**Runmin Dong**
Sun Yat-sen University

**Juepeng Zheng**
Sun Yat-sen University

**Haohuan Fu** [*]
Tsinghua University, Tsinghua SIGS

{chenmx21, zouzh24, zhang-jx22}@mails.tsinghua.edu.cn
ligw8@mail2.sysu.edu.cn; fangwang@cma.gov.cn
{dongrm3, zhengjp8}@mail.sysu.edu.cn; haohuan@tsinghua.edu.cn

## Abstract

Seasonal-scale climate prediction plays a critical role in supporting agricultural planning, disaster prevention, and long-term decision making. In particular, reliable forecasts issued 1-6 months in advance are essential for early warning of flood and drought risks associated with precipitation during the East Asian summer monsoon season. However, while the use of machine learning techniques has advanced rapidly in weather and subseasonal-to-seasonal forecasting, partly driven by the availability of benchmark datasets, their application to seasonal-scale prediction remains limited. Existing seasonal prediction primarily relies on ensemble forecasts from numerical models, which, while physically grounded, are subject to biases and uncertainties at long lead times. Motivated by these challenges, we propose SeasonBench-EA, a benchmark dataset for seasonal prediction in East Asia region. It features multi-resolution, multi-source data with both regional and global coverage, integrating ERA5 reanalysis data and ensemble forecasts from multiple leading forecast centers. Beyond key atmospheric fields, the dataset also includes boundary-related variables, such as ocean state, soil and solar radiation, that are essential for capturing seasonal-scale atmospheric variability. Two tasks are defined and evaluated: 1) machine learning-based seasonal prediction using ERA5 reanalysis, and 2) post-processing of seasonal forecasts from numerical model ensembles. A suite of deterministic and probabilistic metrics is provided for tasks evaluation, along with a hindcast assessment focused on precipitation during the East Asian summer monsoon, aligned with model evaluation protocols used in operations. By offering a unified data and evaluation framework, SeasonBench-EA aims to promote the development and application of data-driven methods for seasonal prediction, a challenging yet highly impactful task with board implications for society and public well-being. Our benchmark is available at https://github.com/SauryChen/SeasonBench-EA.

---

[*]Corresponding author

39th Conference on Neural Information Processing Systems (NeurIPS 2025) Track on Datasets and Benchmarks.

# 1 Introduction

Seasonal-scale climate prediction plays a critical role in various socioeconomic activities, including agricultural planning, disaster prevention and reduction, and water resource management. As extreme weather and climate events become more frequent under climate change, there is growing demand for accurate and reliable forecasts across multiple timescales, spanning from weather to subseasonal and seasonal predictions [1, 2, 3, 4]. In the domains of weather and subseasonal-to-seasonal (S2S) forecasting, recent progress has been supported by the availability of public datasets and benchmarks [5, 6, 7], which have accelerated the development of machine-learning models. Some of these models have demonstrated performance exceeding that of long-established earth system models [8, 9, 10]. In contrast, seasonal prediction has received relatively less attention within the field of artificial intelligence. Machine-learning models specifically designed for this task remains limited, partly due to the lack of standardized datasets and benchmarks.

Different from weather (1-15 days) and S2S (15-45 days) forecasting, seasonal targets lead time of 1–6 months, with a focus on accurately capturing monthly-mean climate states and their anomalies relative to climatology. While weather forecasting is highly sensitive to initial conditions [11], seasonal prediction depends predominantly on slowly varying boundary conditions, such as ocean states, sea ice coverage, soil temperature, and solar radiation, which regulate long-term atmospheric dynamics. Moreover, in contrast to S2S forecast that focus on the evolution of short-term disturbances [12], seasonal prediction emphasizes monthly variability and deviations from climatology normals. These fundamental differences pose unique challenges for the development and evaluation of seasonal prediction models.

To bridge this gap, we introduce SeasonBench-EA, a multi-resolution, multi-source benchmark dataset focused on seasonal prediction, with an emphasis on the East Asia region, as described in Figure 1. East Asia presents unique challenges for seasonal forecasting due to its complex monsoon systems, strong ocean-atmosphere interactions, and highly uneven spatiotemporal distribution of precipitation. These characteristics demand models that can capture regional-specific dynamics, which are often underrepresented in global-scale approaches. However, current AI-based forecasting models have been developed at global-scale [8, 13, 14], with limited exploration of regional solutions. Considering the critical role of boundary conditions in seasonal prediction and the need for regional-specific modeling, as different regions are influenced by distinct climate conditions, SeasonBench-EA provides 0.25° resolution data over East Asia (58-163°E, 8-60°N) and 1° resolution data globally. This design captures fine-scale regional features while preserving global boundary information, enabling more accurate regional forecasts under reasonable computational costs. The data configuration is also inspired by the nested-grid approach commonly used in regional numerical models, and may facilitate the development of nested architectures in AI-based seasonal prediction [15, 16].

With the collected data, we benchmark two practical tasks with various representative data-driven models to evaluate seasonal prediction in a systematic manner: 1) machine learning-based prediction

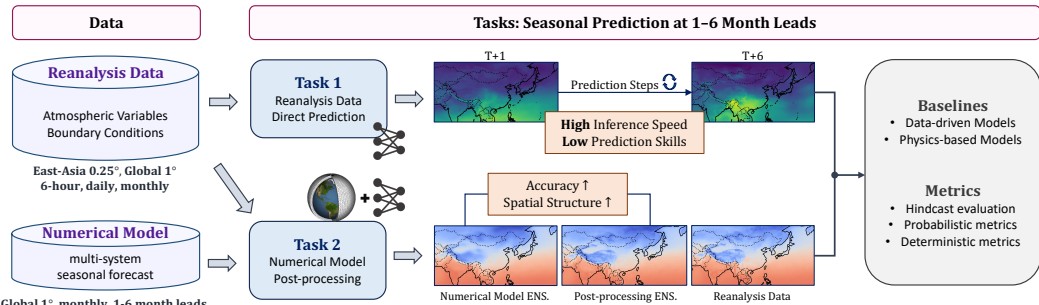

Figure 1: Overview of SeasonBench-EA, a multi-resolution, multi-source benchmark dataset designed for seasonal prediction in East Asia. It integrates ERA5 reanalysis data, including atmospheric variables and key boundary conditions, as well as ensemble seasonal forecast results from leading operational centers. SeasonBench-EA supports two tasks: 1) machine learning–based prediction from reanalysis, and 2) post-processing of the numerical model ensemble outcomes. In addition to standard deterministic and probabilistic metrics, it also provides a hindcast evaluation for assessing model's long-term predictive skill and robustness.

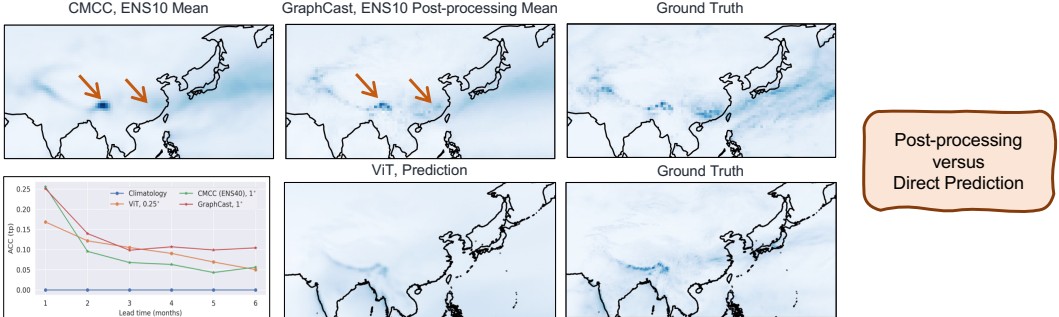

Figure 2: Overview of the performance for the two benchmark tasks at a two-month lead time, using ViT for direct prediction and GraphCast for ensemble post-processing as examples.

using ERA5 reanalysis, and 2) post-processing of numerical model ensemble forecasts. While the former directly leverages historical data, it often suffers from the blurring effect and substantial uncertainty at extended lead times [7], limiting its ability to capture sharp spatial structures and extreme events. In contrast, post-processing offers a hybrid strategy that combines physically informed guidance with data-driven corrections, which can mitigate some of these limitations (orange arrows in Figure 2). However, its effectiveness remains highly dependent on the quality of the ensemble forecasts. Moreover, the anomaly correlation coefficient (ACC) scores across both tasks underscore the current limitations and highlight the challenges faced in the seasonal prediction task.

## 2 Related Work

In recent years, a growing number of benchmark datasets have been introduced to support the development of data-driven models across various forecasting timescales, including weather [5, 17, 6], S2S [18, 19, 7], and long-term climate projection [20, 21]. Besides, datasets have also been proposed for related applications such as statistical downscaling [22] and extreme weather detection [23, 24]. A summary of representative datasets focused on prediction is provided in Table 1.

**Gap at the seasonal timescale.** While weather and climate benchmarks have advanced machine learning for medium-range weather forecasting and long-term climate projections[25, 26, 13, 8, 27, 9], a clear gap remains at the seasonal timescale. At this timescale, machine learning models exhibit blurring effects and struggle with boundary condition signals, while numerical models suffer from systematic drift and model bias.

**Limited variable diversity for physical consistency.** Most existing datasets beyond the weather scale focus on surface variables like temperature and precipitation [19, 18, 21, 20]. However, variables such as geopotential and specific humidity are essential for capturing circulation patterns. For instance,

Table 1: Comparison of SeasonBench-EA with existing datasets for AI-based prediction. SeasonBench-EA fills the gap between medium-range weather forecasting and long-term climate projection by supporting both prediction and post-processing with multiple variables. Note: H = Hour, D = Day, W = Week, M = Month, Y = Year. ✗ in Multi-Var means that only temperature and precipitation are included.

| Benchmark | Time Res./ Lead Time | Spatial Res./ Region | Reanalysis / NWP Data | Prediction/ Post-processing | Multi-Var |
|---|---|---|---|---|---|
| ENS-10 [17] | 24H / 48H | 0.5° global | ✓ / ✓ | ✗ / ✓ | ✓ |
| WeatherBench2 [6] | 6H / 15D | 0.25° global | ✓ / ✓ | ✓ / ✗ | ✓ |
| ChaosBench [7] | 1D / 44D | 1.5° global | ✓ / ✓ | ✓ / ✗ | ✓ |
| SubseasonalClimateUSA [19] | 2W / 6W | 1° contiguous U.S | ✓ / ✓ | ✓ / ✓ | ✗ |
| SubseasonalRodeo [18] | 2W / 6W | 1° western U.S | ✓ / ✓ | ✓ / ✗ | ✗ |
| **SeasonBench-EA** | 1M / 6M | 0.25° EA & 1° global | ✓ / ✓ | ✓ / ✓ | ✓ |
| ClimateSet [21] | 1M / 251Y | 250km global | ✗ / ✓ | ✓ / ✗ | ✗ |
| ClimateBench [20] | 1Y / 500Y | 250km global | ✗ / ✓ | ✓ / ✗ | ✗ |

Table 2: Reanalysis variables included in SeasonBench-EA. Variables highlighted in brown are available in the reanalysis dataset, while the others are included in both the reanalysis and the numerical model ensembles.

| Type | Variables |
|---|---|
| surface | 2m temperature, mean sea level pressure, total precipitation |
| pressure @ 1000, 850, 700, 500 200 hPa | temperature, u/v component of wind, geopotential, specific humidity |
| boundary | boundary layer height, surface solar radiation downwards, soil temperature, volumetric soil water layers, snow albedo, snow depth, sea surface temperature, sea ice cover |
| constant | geopotential at surface, land sea mask, soil type |

experts often interpret geopotential anomalies to infer likely precipitation distributions, helping ensure the physical consistency of seasonal forecasts.

**Seasonal prediction in East Asia.** Machine learning has shown promise in seasonal prediction and model correction tasks over East Asia, especially for precipitation prediction [28, 29, 30, 31, 32]. However, existing studies are often developed for specific regions and variable sets, and adopt inconsistent evaluation protocols. Currently, no publicly available dataset supports seasonal prediction over East Asia with standardized evaluation settings, making it difficult to systematically compare methods and limiting broader participation.

# 3   SeasonBench-EA

SeasonBench-EA integrates ERA5 reanalysis data [33, 34, 35, 36] and ensemble forecasts [37, 38] from leading operational centers. Given that seasonal prediction is sensitive to boundary conditions, dataset covers essential atmospheric variables and critical boundary layer variables such as ocean, soil, and solar radiation, supporting robust modeling of long-term climate conditions. While evaluation is conducted at the monthly scale, aligning with the typical temporal resolution of seasonal prediction, the reanalysis data is available at hourly, daily, and monthly resolutions, and the ensemble forecasts are provided at monthly resolution, facilitating flexible training and assessment across temporal scales.

## 3.1   Reanalysis Data

Reanalysis data serve as input variables for the seasonal prediction task and the ground truth for numerical model post-processing. The selected variables are summarized in Table 2. Variable selection follows two criteria: the ability to characterize the large-scale atmospheric circulation, and well-established relevance to seasonal precipitation prediction [28, 31, 32, 39, 40, 41, 42], which is a key focus of seasonal prediction in the East Asian monsoon region. For example, sea surface temperature and sea ice cover influence teleconnections, while snow depth and albedo over the Tibetan Plateau affect moisture transport. The reanalysis data spans from 1940 to 2024 at monthly resolution, and from 1991 to 2024 at 6-hourly and daily resolutions. The total volume of the reanalysis data is approximately 715 GB.

## 3.2   Seasonal Forecasts from Numerical Model Ensembles

The variables used in numerical model ensemble are a subset of those in the reanalysis dataset, as summarized in Table 2, selected to support the ensemble post-processing task. The selection is also based on the variables' availability and temporal coverage across different forecasting systems. For instance, certain systems lack key variables: the JMA model does not provide variables at 1000 hPa, while the UK Met Office, NCEP and BOM models do not include variables such as snow depth, therefore omitted in the dataset. Additionally, we prioritize system versions of each numerical model that cover all calendar months and span longer temporal coverage. When multiple system versions are available for a numerical model, we choose the latest version available at the time of data download.

Table 3: Summary of numerical model ensemble systems included in SeasonBench-EA.

| Center | CMCC | DWD | ECCC | ECMWF | Meteo-France |
|---|---|---|---|---|---|
| System | SPS 3.5 | GCFS2.1 | GEM5-NEMO | SEAS5 | System 8 |
| Ensemble members | 40 | 30 | 10 | 25 | 25 |

Seasonal prediction data from five operational centers are included, as summarized in Table 3, with additional details for each system provided in Supplementary Section A.

All ensemble forecasts provide global coverage at a spatial resolution of $1°$, with a monthly temporal resolution, consistent with the global reanalysis data. The dataset spans the period from 1993 to 2024, with a total data volume of approximately 1.3 TB for the multi-model ensemble component. The data processing focused on cropping the data to the East Asia region, aligning the spatial and temporal grids between the numerical model ensembles and reanalysis data, as well as performing unit conversion and normalization.

## 3.3 Baselines

SeasonBench-EA supports two tasks: 1) seasonal prediction based on reanalysis data, and 2) post-processing of seasonal forecasts from numerical model ensembles. While our baseline focus on a commonly used set of target variables (*t2m, tp, t_850, z_500, q_700*), the benchmark remains flexible, allowing users to define custom variable combinations for specific research needs.

We construct separate baselines for the two tasks. For the seasonal prediction task, models are trained on reanalysis data within the East Asia region, aligning with the goal of regional forecasting. For the post-processing task, models are built on global scale, but evaluated over the East Asia domain to assess improvements in regional skill.

SeasonBench-EA includes a variety of representative data-driven architectures for both tasks: U-Net [43], ViT [44], FNO [45], and VAE [46]. For the post-processing, we additionally include architectures designed for global-scale modeling, including the SFNO [14] and GraphCast [8]. Besides, the monthly climatology and persistence predictions are used as two physics baselines, following [5, 22, 7]. These baselines are distinguished from data-driven models because they rely solely on climatological statistics. A description of these model configuration is provided in Supplementary Section F.

## 3.4 Metrics

SeasonBench-EA provides both deterministic and probabilistic metrics that are commonly used in seasonal prediction and ensemble forecast. For deterministic metrics, we include root mean square error (RMSE), bias, Willmott's index of agreement (WI), anomaly correlation coefficient (ACC), energy spectrum, and critical success index (CSI). For probabilistic metrics, we adopt rank histogram, continuous ranked probability score (CRPS), and spread–skill ratio (SSR). A detailed description of these metrics is provided in Supplementary Section B.

## 3.5 Hindcast Evaluation

Hindcast evaluation provides a retrospective framework for validating predictive models by comparing their forecasts anomalies with historical observation anomalies. This approach is widely adopted in operations to assess model performance across multiple years and different initial time. In SeasonBench-EA, we employ two evaluation metrics:

**Anomaly Correlation Coefficient (ACC_hindcast)** Unlike the standard ACC metric, ACC_hindcast (Eq. 1) further removes the climatological mean specific to each data source, *i.e.* forecast climatology for predictions and observation climatology for ground truth. This adjustment helps correct for the model's systematic biases and enables a more robust and accurate evaluation of a model's ability to

capture interseasonal variability.

$$ACC = \frac{\sum_{i=1}^{H \times W} \left( \Delta f - \overline{\Delta f} \right) \left( \Delta O - \overline{\Delta O} \right)}{\sqrt{\sum_{i=1}^{H \times W} \left( \Delta f - \overline{\Delta f} \right)^2 \sum_{i=1}^{H \times W} \left( \Delta O - \overline{\Delta O} \right)^2}}, \Delta f = f - \overline{f}, \Delta O = O - \overline{O}, \quad (1)$$

where $f$ and $O$ represent the forecast and observe values at each grid point, $\overline{f}$ and $\overline{O}$ represent their climatological means over the evaluation years. The terms $\Delta f$ and $\Delta O$ are the corresponding anomalies. $H$ and $W$ indicate the number of latitude and longitude grid points, respectively.

**Temporal Correlation Coefficient (TCC)** TCC (Eq. 2) is calculated at each spatial grid to evaluate the temporal consistency between forecasts and observations anomalies over multiple years. It reflects a model's capacity to reproduce interannual variability for the target month at local scales, which is essential for skillful seasonal prediction.

$$\text{TCC}i, j = \frac{\sum_{t=1}^{T}(f_t^{(i,j)} - \bar{f}^{(i,j)})(O_t^{(i,j)} - \bar{O}^{(i,j)})}{\sqrt{\sum_{t=1}^{T}(f_t^{(i,j)} - \bar{f}^{(i,j)})^2}\sqrt{\sum_{t=1}^{T}(O_t^{(i,j)} - \bar{O}^{(i,j)})^2}}, \quad (2)$$

where $f_t^{(i,j)}$ and $O_t^{(i,j)}$ are the forecast and observe values at grid point $(i, j)$ for year $t$, while $\bar{f}^{(i,j)}$ and $\bar{O}^{(i,j)}$ are their corresponding temporal means at that grid point over the evaluation period of $T$ years.

## 4    Results and Analysis

In this section, we present the results for the following variables: total precipitation (tp), 2-meter temperature (t2m), temperature at 850 hPa (t_850), geopotential at 500 hPa (z_500), and specific humidity at 700 hPa (q_700). These variables are critical for describing large-scale atmospheric circulation, with total precipitation serving as a core predictand in seasonal forecasting applications.

### 4.1    Prediction

For the seasonal prediction task, we adopt an auto-regressive forecasting strategy. Models are trained using reanalysis data from 1940 to 2015, validated on 2016 to 2019, and evaluated over 2020 to 2024. Monthly climatology for anomaly computation is derived from the 1991-2020 reference period. The years 2020 to 2024 are selected for testing due to their diverse seasonal precipitation patterns over East Asia, allowing for evaluation under a range of climate conditions (see Supplementary Section C). Additional experiments, including a simple linear regression model, rolling-window evaluations to assess temporal robustness, multi-seed training to evaluate model stability, and detailed results for the seasonal prediction task, are provided in Supplementary Section D.

**Loss of predictive skills relative to climatology.** As shown in Figure 3 (a) and (b), all models exhibit higher RMSE compared to the monthly climatology baseline, with ACC values even dropping below zero at several lead months. This consistent performance degradation across different architectures suggests that the limitation is not specific to any particular model design, but rather stems from fundamental challenges in seasonal prediction. In particular, the baselines lack sensitivity to boundary-driven signals such as solar radiation and sea surface temperature, which are critical at seasonal timescales. Moreover, they fail to incorporate the broader environmental context, where global or surrounding regional conditions serve as essential boundary constraints for local climate evolution. These findings highlight the need to develop models that are physically informed, aware of boundary conditions, and capable of capturing long-range spatiotemporal dependencies.

**Lack of small-scale variability in model predictions.** The energy spectrum plots depicted in Figure 3 (d) show that models exhibit a significant reduction in spectral amplitude at high wavenumbers compared to the reanalysis, particularly for tp. This indicates a substantial loss of small-scale variability in the predictions. At a six-month lead time, the predicted tp fails to reproduce detailed spatial structures and deviates from the climatology patterns (Figure 3 (c)). In addition, all models except FNO show spurious peaks at certain small-scale wavenumbers, which may reflect the instability in predicting fine-scale processes. Potential contributing factors include the lack of physical constraints, error accumulation in auto-regressive inference, and the limited capacity of pixel-level loss functions to penalize spatial discontinuities.

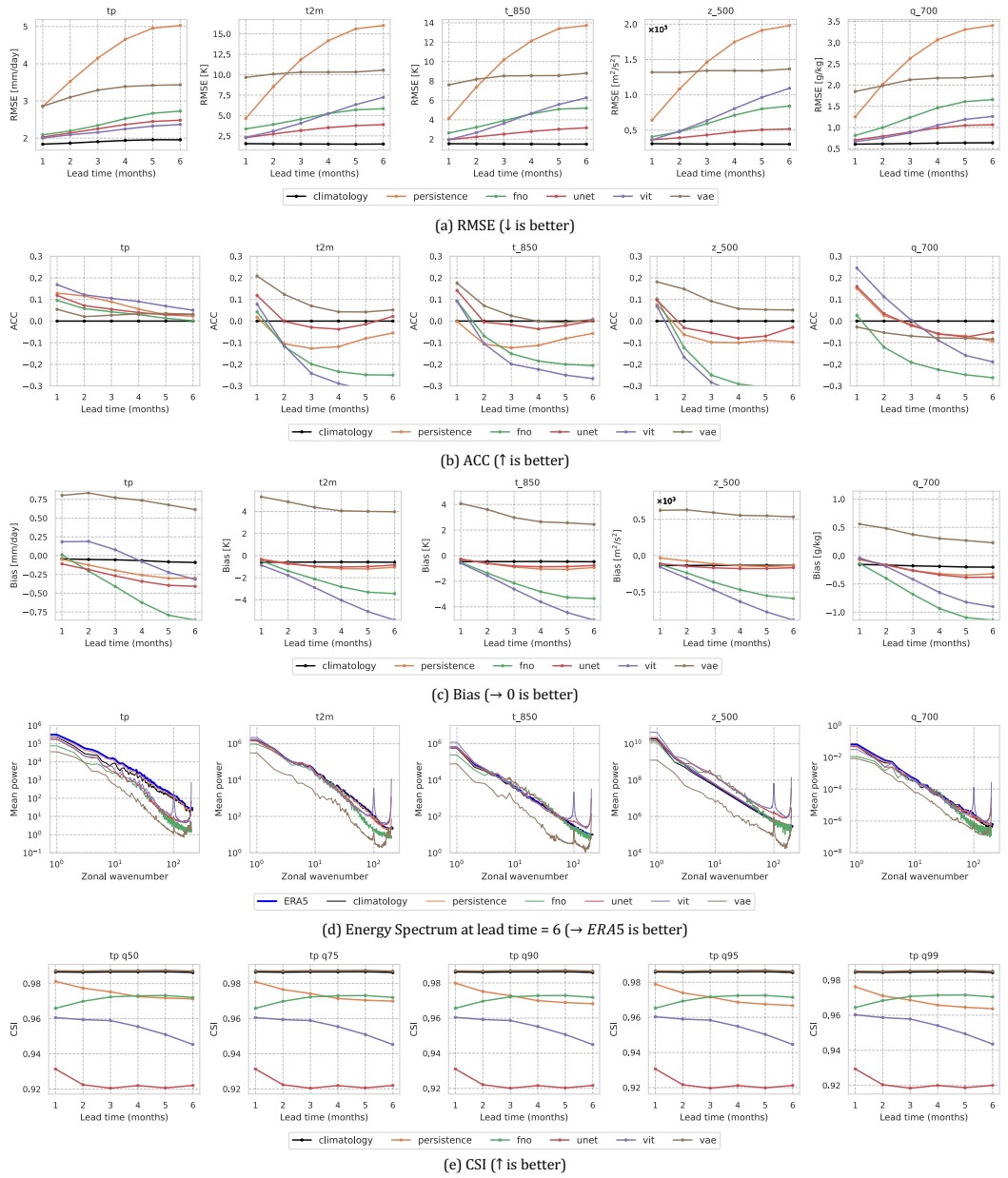

Figure 3: Evaluation of seasonal prediction performance using deterministic metrics.

**Limited benefits from longer autoregressive steps.** In medium-range weather forecasting and S2S prediction, training models with longer autoregressive steps has been shown to improve stability and performance by better capturing temporal dependencies [7]. However, extended steps fails to enhance model performance in this task. As shown in Figure 4, model performance does not improve monotonically with longer training steps. The growing uncertainty and weakened signal-to-noise ratio at long lead times could limit the effectiveness of autoregressive learning strategies. Further details are presented in Supplementary Section D.

## 4.2   Post-Processing

For the post-processing of seasonal forecasts from numerical model ensembles, models are trained to directly output corrected variable fields at all lead times. Training is conducted on global-scale data to incorporate large-scale boundary information, while evaluation is performed over East Asia

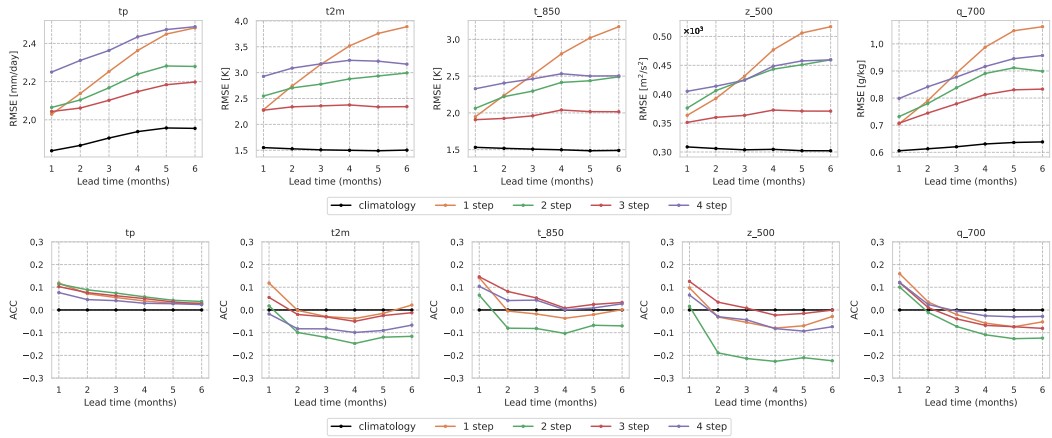

Figure 4: Performance comparison of U-Net models trained with different autoregressive steps.

to ensure consistency with the prediction task setup. Specifically, models are trained on data from 1993 to 2024, excluding the validation (2009–2011) and test (2013–2016) periods. The test period is chosen to accommodate inconsistencies in data coverage, as several numerical models have missing years in their forecast records. The monthly climatology is computed using the 1991–2020 period from reanalysis data with a global resolution of 1°. Figure 5 shows the post-processing results using ensemble forecasts from CMCC as an example. Following [17], SeasonBench-EA uses the first 10 ensemble members. Additional evaluation results for post-processing, multi-seed training to assess model stability, and GraphCast-based results for ECMWF are listed in Supplementary Section E.

**Numerical model post-processing improves forecasting skills.** Post-processed results based on 10 ensemble members outperform the original 40-member ensemble in both RMSE and ACC, demonstrating that data-driven correction can effectively improve forecast accuracy. Notably, the post-processed forecasts also surpass the direct prediction models, highlighting the advantage of incorporating numerical model guidance and ensemble diversity to enhance seasonal prediction. From the perspective of energy spectrum, spurious peaks at small scales are significantly reduced, suggesting improved physical consistency and spatial coherence. Despite the improvements, ACC values remain below the threshold of skillful prediction, indicating that the post-processed forecasts still fall short in capturing accurate anomaly signals, particularly at longer lead times.

**Precipitation correction remains challenging.** Among all target variables, the improvement from post-processing is limited for total precipitation. As shown in Figure 5 (c), the corrected forecasts exhibit a significant drop in amplitude at high wavenumbers, indicating a failure to recover fine-scale spatial variability. Additionally, Figure 5 (e) shows that the rank histogram for total precipitation retains a U-shaped pattern after correction, suggesting that the ensemble remains underdispersive. Notably, the climatology baseline continues to outperform the corrected forecasts, especially for precipitation. During training, models optimized with RMSE-based objectives are likely to converge to average, leading to smooth predictions and the loss of spatial details. These results highlight the need for improved model architectures, such as GraphCast, and loss functions that better preserve spatial details and capture precipitation variability.

### 4.3 Hindcast

We further perform a hindcast evaluation on three representative cases: 1) direct prediction using the ViT model, 2) the ensemble mean of the first 10 members from CMCC, and 3) the ensemble forecasts post-processed by GraphCast. The evaluation targets total precipitation during the summer season (June–August), with all forecasts initialized in March. To assess both spatial and temporal forecast skills, we report $ACC_{hindcast}$ and TCC. The hindcast period spans 2006-2020, while 2021–2024 is used for validation. All remaining years are included in the training set. Both the GraphCast and ViT models are trained with four random seeds to compute the mean and standard deviation of their hindcast performance. The hindcast results for total precipitation during the summer season are demonstrated in Figure 6.

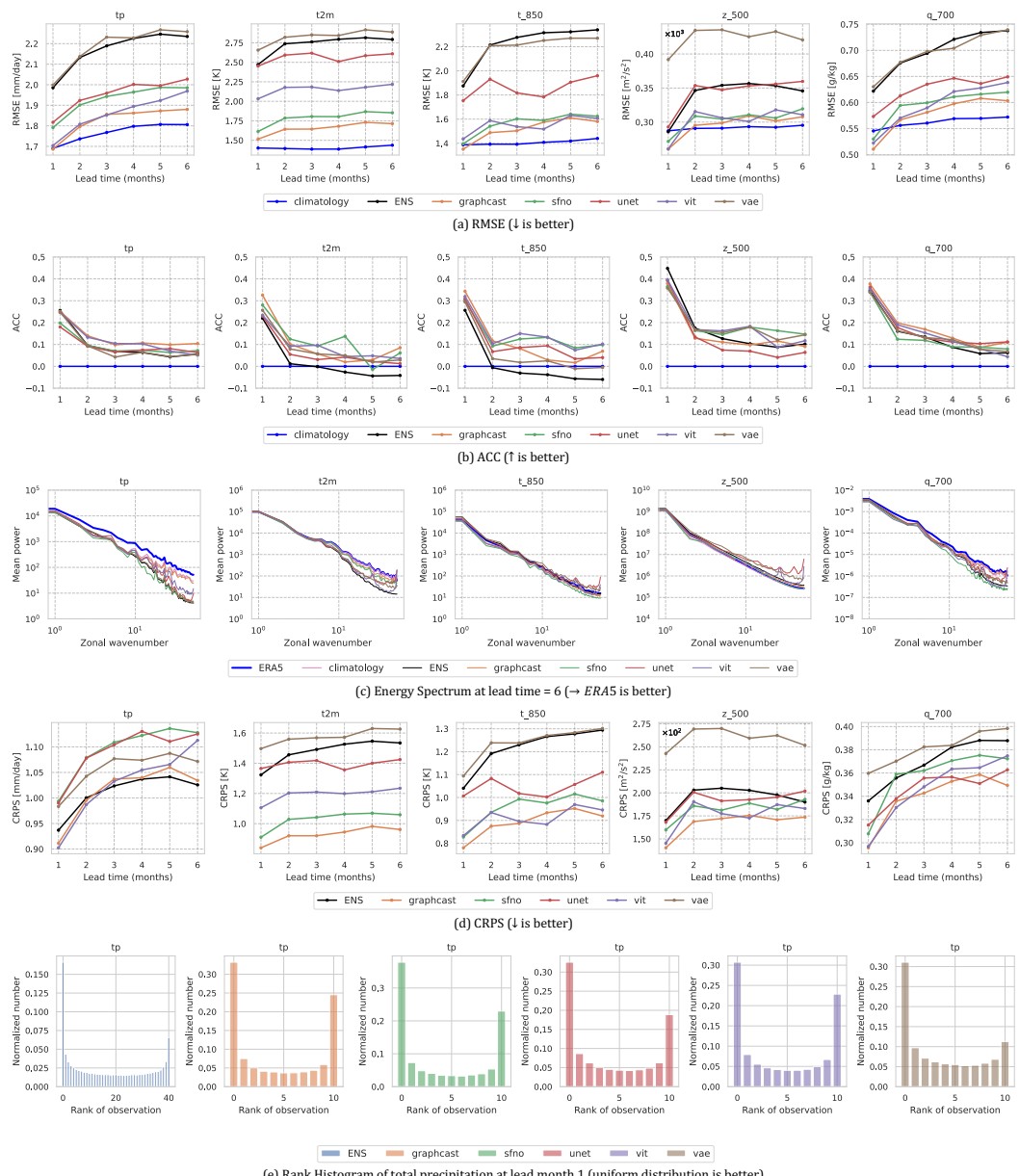

Figure 5: Post-processing results for the ensemble forecasts. "ENS" refers to the original 40-member ensemble forecasts from CMCC, while the other models apply correction using 10 ensemble members.

**Physically informed models outperform data-driven ones, but challenges persist.** Although all ACC values fluctuate around zero, with the value of 0.032±0.131 (ViT), 0.129±0.122 (GraphCast), and 0.119 (numerical ensemble), physically informed methods, such as GraphCast-based post-processing and numerical ensemble forecasts, outperform the purely data-driven ViT model. This highlights the benefit of incorporating physical priors or leveraging ensemble diversity to improve predictive skill at seasonal timescales. However, these gains in spatial accuracy do not necessarily translate into better temporal consistency. Despite achieving higher ACC values, post-processed forecasts exhibit a decline in TCC scores compared to raw ensembles, suggesting limited preservation of interannual coherence. This may originate from the strong dependence of post-processing models on numerical inputs, and the fact that commonly used loss functions primarily emphasize spatial accuracy while overlooking year-to-year variability.

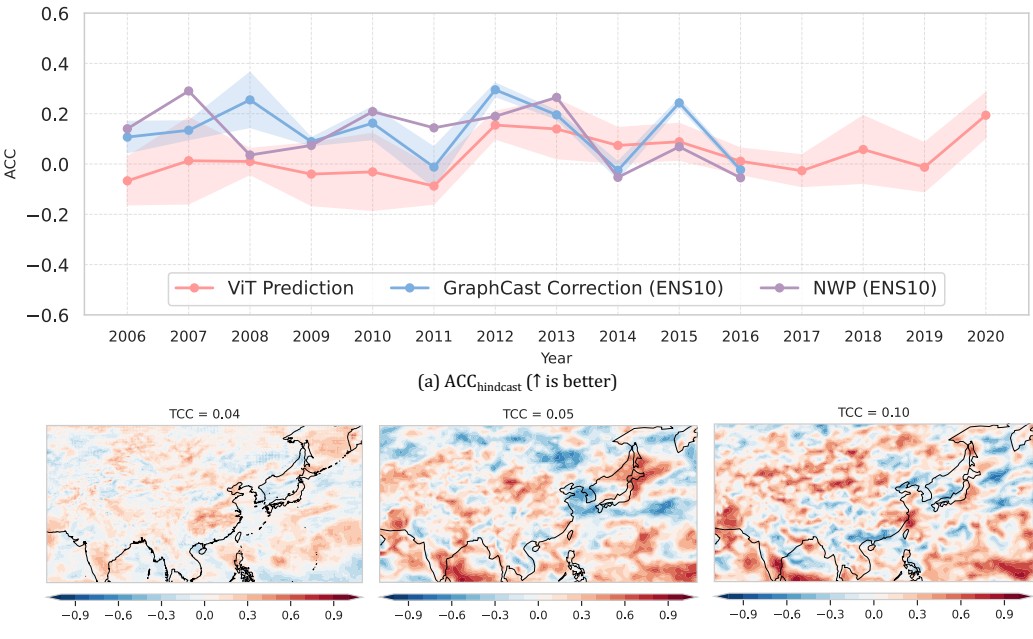

(a) ACC$_{\text{hindcast}}$ (↑ is better)

(b) TCC (↑ is better), from left to right: ViT prediction, GraphCast correction and NWP with 10 ensemble members.

Figure 6: Hindcast results for total precipitation during the summer season from 2006 to 2020, with March as the initialization month. Note that CMCC SPS3.5 lacks data from 2017 to 2020, leading to missing values in those years.

## 5 Conclusion

In this work, we present SeasonBench-EA, a multi-resolution, multi-source benchmark dataset designed to advance data-driven research in season prediction. By integrating ERA5 reanalysis and numerical ensemble forecasts from leading centers, SeasonBench-EA provides a unified framework to evaluate both direct prediction and post-processing tasks. Our benchmark across a range of representative models reveal that the performance remains limited, especially for key variables like precipitation. Directed prediction models can reproduce large-scale variability but lose skill rapidly with increasing lead time, showing oversmoothed and low-variance forecasts due to the lack of explicit physical constraints. Post-processing models, leverage ensemble diversity to enhance spatial accuracy and reduce small-scale noise, yet remain limited by biases in numerical forecasts and weakened temporal coherence. Overall, physically informed methods outperform purely data-driven ones at the current stage, underscoring the importance of boundary information and physical priors. It demands further methodological innovations that can effectively incorporate physical constraints, capture boundary-driven variability, and model long-range spatiotemporal dependencies.

Several limitations remain in SeasonBench-EA. Due to computational constraints, not all available numerical forecast models are evaluated (additional results for ECMWF are provided in Section E). Also, while current dataset includes key atmospheric and boundary variables, incorporating additional boundary conditions, such as sea surface salinity and subsurface ocean temperatures, could further improve the representation of long-term drivers of seasonal variability. We provide data download script so that users can add variables to the dataset for their specific research need. SeasonBench-EA will be continuously updated to include more data sources, variables, and evaluation protocols.

Beyond the core prediction tasks, the dataset's multi-resolution design also enables downscaling applications and supports the development of nested model architectures, similar to the grid nesting strategies commonly employed in regional numerical weather and climate models. Although nested data-driven models have not yet been benchmarked in this work, the dataset provides a solid foundation for the future exploration. Such architectures allow for high-resolution regional predictions that incorporate broader-scale boundary conditions from global contexts, while maintaining reasonable computational costs. This is particularly valuable in regions like East Asia, where complex land–ocean–atmosphere interactions demand both local precision and global awareness.

## Acknowledgments and Disclosure of Funding

The work is supported by the National Key Research and Development Plan of China (Grant 2023YFB3002400) and National Natural Science Foundation of China (Grant T2125006).

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
