# OpenReview forum: "SeasonBench-EA: A Multi-Source Benchmark for Seasonal Prediction and Numerical Model Post-Processing in East Asia"
_NeurIPS.cc/2025/Datasets_and_Benchmarks_Track — NeurIPS 2025 Datasets and Benchmarks Track poster_

### Official Review · Reviewer_fQ8y · 2025-06-25

**Rating:** 4
**Confidence:** 4

**Summary:**

The manuscript introduces SeasonBench-EA, a dataset and benchmark that tackles seasonal weather prediction on lead times of one to six months. The dataset offers two tasks: Pure weather prediction with data-driven models and post-processing of outputs from numerical weather prediction models. A reasonable selection of data-driven models underlines the challenges of seasonal prediction and justifies the installation of a seasonal benchmark.

**Additional Feedback:**

### Questions
- Table 2: Why is `geopotential at surface` listed as a `constant`? I would expect this variable to appear in the `surface` line.
- Several studies report that GraphCast is unstable when generating forecasts beyond four weeks [1](https://agupubs.onlinelibrary.wiley.com/doi/full/10.1029/2023MS004021). How did you get forecasts out to six months with GraphCast? I would appreciate supporting information in the manuscript.
- Do you have results for post-processing of other NWP models beyond CMCC? It would be very interesting to see the relative improvement of each model and the final performances.

### Typos and Suggestions
- L97: our dataset? (add "our" or "this")
- Figure 4: change "step n" in legend to "n steps" to clarify that models were trained with $n$ autoregressive steps. I was first confused, since I typically relate a "step" to a forecast step, i.e., association to lead time.
- An approachable visual comparison between "prediction" and "post-processing" results would be highly appreciated to conveniently assess the differences between these two tasks.

**Dataset Code Accessibility:**

No

**Dataset Code Comments:**

1. I could not access the baselines behind the [SeasonBench-EA (subset)](https://dataverse.harvard.edu/dataset.xhtml?persistentId=doi:10.7910/DVN/EPEUGO) link in the repository.
2. Downloading the full dataset following [the other link](https://pan.baidu.com/s/1p78We3pCwU-eF3Xp-I0-Uw) provided in the repository was difficult, as the language of the portal is Chinese. Following my intuition, I eventually downloaded a `BaiduNetdisk_mac_4.49.1_x64.dmg` file, which I did not want to install on my computer.
3. Browsing through the repository, it seems to provide sufficient information to train and evaluate models. To get started, yet, some instructions about how to install the dataset package or which python packages are need would be helpful.

**Ethical Considerations:**

No, there are no or only very minor ethics concerns

**Final Justification:**

After reading the rebuttal and downloading parts of the dataset, my concerns are widely resolved and I have increased my score from 2 to 4.

**Limitations Weaknesses:**

1. Floating environments (figures and tables) need much better formatting. E.g., Figure 1 hardly stands out from the text and Table 1 exceeds the paper margin. Also, I recommend to position floating environments at the top or bottom of the pages.
2. The selection of variables appears reasonable, but a thorough explanation of why these particular variables were chosen is missing. It would be essential to understand the motivation to include each variable and in what way each variable is expected to contribute to seasonal prediction. Also, as mentioned in the conclusion (I very much appreciate this argument!), the selected variables might not cary the necessary information and signals to allow data-driven models excell in seasonal prediction. Appart from subsurface ocean temperature and salinity, I'm thinking of variables characterizing the land surface, e.g., vegetation indices, such as leaf area index (LAI) or NDVI, soil properties, and surface cover (cf. corine land cover).
3. The numerical models in Table 3 are neither introduced in the manuscript, nor can I find details where their forecasts come from originally and why each system was chosen. For example, why did authors select GEM5-NEMO from ECCC or GCFS2.1 from DWD? (Edit: I found some information in the appendix. Please refer to this section in the manuscript and add reasons for choosing each model and version).
4. Claim in ll 184-186 needs substantiation. Please mention studies that suggested an improvement of S2S prediction performance whe increasing the number of autoregressive steps in training.
5. Training curriculum is intransparent. Please add information about number of epochs/iterations, optimizer and learning rate, schedulers, and other setting employed in the model training. Also, the hardware and training times of different models (along with memory consumption and parameter count) would be of value for readers. (Edit: found the information in the supplementary material. Please add a reference directing the reader in Section 3.3 to Appendix F.) Seeing the different numbers of parameters, how did authors decide on the configuration of the architectures? Did you perform an exhaustive grid search to find the best layer/neuron/activation/optimizer/etc. configurations or are the reported models the only setting that was executed?
6. Single models only allow limited conclusions. It would be helpful if several models were trained with different seeds to quantify mean and standard deviation of different runs. This should be feasible given four models in the first task (prediction) and five models in the second task (post-processing).
7. I did not quite understand how the nesting was implemented for the data-driven models. Please add information about the treatment of boundary conditions from global 1° to regional 0.25° resolution.
8. Section 4.3 tackles an interesting aspect. However, given the results come from single shot models without depicting standard deviations, the conclusion about superior NWP models seems daring. Authors may train several GraphCast and ViT models with different seeds and bootstrap different ensemble members from CMCC to come up with statistics that might allow a more substantiated conclusion.
9. A concrete leaderboard and a goal what models should be capable of would be helpful.
10. My major concern, which leads to my decision to reject the introduced dataset in its current form, relates to the limited data availability and accessibility (see according sections below).

**Strengths Contributions:**

The manuscript tackles a gap in weather prediction by focusing on a forecast range of 1-6 months. Improving model forecast on these ranges is crucial for society under various aspects. The limitations of seasonal prediction with data-driven models is well supported in Figure 3 (b).

Data-driven baselines are chosen carefully to cover various types of backbone architectures used in deep learning weather prediction. Additional arguments for what reason each architecture was chosen would be appreciated (adding Pangu-Weather as a transformer architecture would be interesting too).

Similarly, the evaluation metrics cover a good range to quantitatively compare the quality of different forecasts. It would be helpful, though, to add an explanation what each metric represents, that is, what does it mean, if a model performs particularly good/bad on a specific metric?

---

> ### Author Rebuttal · Authors · 2025-07-31
>
> # Author response to Reviewer fQ8y
> We sincerely thank Reviewer fQ8y for the insightful and valuable comments.
> ## Q1: Formatting
> We have carefully revised the formatting issues in the paper. The floating environments will be repositioned to the top or bottom of the pages as recommended.
> ## Q2: Variable selection
> The variables in SeasonBench-EA are selected based on two main criteria:
> 1. Their ability to represent the large-scale atmospheric circulation, such as geopotential and temperature, which are key predictors for weather to seasonal forecasting.
> 2. Their well-established relevance to East Asian climatology, as documented in recent studies [1-3]. For example, SST and sea ice cover influence teleconnection patterns, while snow depth and albedo over the Tibetan Plateau affect moisture transport.
>
> Regarding variables such as NDVI, LAI and surface cover, we fully agree that they are valuable predictors, due to their role in land-atmosphere coupling. We initially exclude these variables due to the *inconsistent temporal and spatial coverage* compared to ERA5 and NWP data.
>
> We include key surface variables such as soil type, which is used by the IFS land surface scheme to represent the classification of soil, as well as volumetric soil water content in multiple layers. In response, we will continuously incorporate land variables from ERA5-land, such as LAI and total evaporation.
>
> Also, to support community use and extensibility, we will provide data download and preprocessing scripts, so that users can add variables to the dataset for their specific research need. We will revise the manuscript to clarify the variable selection criteria and discuss the limitation.
> 1. "Deep learning for seasonal prediction of summer precipitation levels in eastern China."
> 2. "Assessing predictive attribution in NMME forecasts of summer precipitation in eastern china using deep learning."
> 3. "Skillful seasonal predictions of continental East-Asian summer rainfall by integrating its spatio-temporal evolution."
> ## Q3: Numerical model selection
> We will add a reference to Supplementary Section A and explicitly clarify the reason for selecting each system and model. We filter models based on variable availability and temporal coverage. For instance, some systems lack key variables: the JMA model does not provide 1000 hPa data, while the UKMO, NCEP and BOM models do not include variables such as snow depth. Additionally, we prioritize system versions of each numerical model that cover all calendar months and span longer temporal coverage. When multiple system versions are available for a numerical model, we choose the latest version available at the time of data download.
> ## Q4: Reference for autoregressive steps
> We will add a reference to support this claim. Section G.1 in [1] investigates the impacts of autoregressive training steps on S2S predition using U-Net models trained with 1 and 5 iterative steps. As shown in Figure S7 of [1], training with 5 steps yields better performance under vision-based evaluations metrics.
> 1. "Chaosbench: A multi-channel, physics-based benchmark for subseasonal-to-seasonal climate prediction."
> ## Q5: Training curriculum
> We will add a reference to Supp. Section F in Section 3.3.
>
> Regarding model configurations, since most of the baselines are adapted from established architectures in prior research, we do not perform exhaustive grid search. Instead, we first adopt reasonable default settings from previous work, and make some changes (e.g. number of channels, learning rate, activation functions) based on prediction and correction results to ensure stable training and better performance. For example, in FNO, we keep the modes = 16 from its original paper since changing this value does not lead to significant improvements. We use AdamW optimizer across all experiments.
>
> To facilitate comparison between different configurations of the same architecture, we add the *--version* argument in our evaluation and visualization scripts, which allows easy tracking of the performance. This is particularly useful during model development stage.
> ## Q6: Training with different seeds
> Due to time and computational constraints during the rebuttal phase, we conduct multi-seed evaluations for one representative model in each task, i.e. U-Net for prediction and GraphCast for correction. Each model is trained with three additional seeds, resulting in four seeds in total. The results are reported in tables below as mean ± std, with the original value shown in parentheses for comparison. Due to character limitations, ACC values for Z500 and t2m are omitted.
>
> | **UNet** | **Metric** | **1** | **2** | **3**  | **4** | **5** | **6**  |
> | -- | -- | -- | -- | -- | -- | -- | -- |
> | **Z500**  | RMSE  | 353.83 ± 13.63 (363.46) | 384.93 ± 20.16 (392.71) | 422.42 ± 33.43 (431.39) | 468.12 ± 46.01 (476.75) | 503.05 ± 66.96 (505.98) | 530.21 ± 81.58 (516.84) |
> | **t2m**   | RMSE  | 2.27±0.09 (2.28)  | 2.66±0.18 (2.74)   | 3.00±0.28 (3.16)  | 3.29±0.33 (3.52)  | 3.52±0.36 (3.76)  | 3.69±0.37 (3.89)  |
> | **tp**  | RMSE| 2.01±0.01 (2.03) | 2.13±0.04 (2.14)  | 2.24±0.06 (2.25)  | 2.34±0.09 (2.36)  | 2.40±0.08 (2.45)  | 2.44±0.06 (2.48)  |
> |  | ACC  | 0.14±0.02 (0.12) | 0.10±0.02 (0.07)  | 0.09±0.03 (0.05)  | 0.07±0.03 (0.04)  | 0.07±0.03 (0.03)   | 0.06±0.02 (0.03)  |
>
> | **GraphCast** | **Metric** | **1**  | **2**  | **3**   | **4**   | **5**  | **6**  |
> | -- | -- | -- | -- | -- | -- | -- | -- |
> | **Z500**  | RMSE  | 256.05±4.54  (260.36) | 298.32±3.63 (295.30) | 296.98±2.26 (298.57) | 300.60±5.88 (309.21) | 304.01±3.56 (301.68) | 310.78 ± 4.80 (307.79) |
> | **t2m**  | RMSE | 1.46±0.04 (1.52) | 1.62±0.03 (1.64)  | 1.66±0.03 (1.64) | 1.66±0.02 (1.68)  | 1.69±0.04 (1.73)  | 1.73±0.02 (1.71)  |
> | **tp**  | RMSE  | 1.68±0.01 (1.69) | 1.80±0.01 (1.80) | 1.85±0.01 (1.86) | 1.87±0.01 (1.86) | 1.88±0.01 (1.87) | 1.91±0.02 (1.88) |
> |   | ACC  | 0.25±0.00 (0.25)  | 0.12±0.01 (0.14)  | 0.10±0.00 (0.10) | 0.10±0.01 (0.11) | 0.10±0.01 (0.10) | 0.08±0.02 (0.10)  |
>
> We hope this address your concern. We will add the results in the revised paper.
> ## Q7: Nested models
> In our current experiments, we do not implement nested models for data-driven methods. Recent studies based on **graph neural networks** have proposed promising solutions. For example, [1] treats low-resolution global data as non-trainable forcing conditions, and [2] proposes a multi-scale graph that densifies the target region for higher-resolution predictions.  We will cite these works at Line 60 for clarity.
> 1. "Probabilistic weather forecasting with hierarchical graph neural networks."
> 2.  "OneForecast: a universal framework for global and regional weather forecasting."
> ## Q8: Hindcast evaluation
> We retrain GraphCast and ViT models using 3 additional seeds (four seeds in total) to calculate the ACC_hindcast (the table below) and TCC (**0.05 ± 0.01 for GraphCast** and **0.04 ± 0.04 for ViT**) in the manuscript setting. Due to the time limit, we do not perform the bootstrap on different ensemble members. Compared with the first ten ensemble members, the highest TCC are 0.06 and 0.07 for GraphCast and ViT, respectively, lowering than the TCC in NWP. We will add the results to the paper.
>
> | Year (ENS 10 ACC) | Correction  | Prediction |
> | -- | -- | -- |
> | 2006 (0.14) | 0.11±0.07  | -0.07±0.11 |
> | 2007 (0.29)  | 0.13±0.04  | 0.01±0.20  |
> | 2008 (0.04) | 0.25±0.13  | 0.01±0.06  |
> | 2009 (0.07) | 0.09±0.02  | -0.04±0.14 |
> | 2010 (0.21)  | 0.16±0.07  | -0.03±0.18 |
> | 2011 (0.14) | -0.01±0.09 | -0.09±0.08 |
> | 2012 (0.19)  | 0.30±0.03  | 0.15±0.06  |
> | 2013 (0.26) | 0.20±0.02  | 0.14±0.14  |
> | 2014 (-0.05) | -0.03±0.04 | 0.07±0.08  |
> | 2015 (0.07)  | 0.24±0.02  | 0.09±0.08  |
> | 2016 (-0.05) | -0.02±0.01 | 0.01±0.06  |
>
> ## Q9: Leaderboard Summary
> We have added summary tables for two tasks in the GitHub repository README. For each variable and lead time, we highlight the best and second-best performing models. We will also revise the conclusion section to summarize the strengths and limitations of different methods, offering clearer guidance on what future models should aim to achieve.
>
> ## Q10: Dataset availability
> We have uploaded all baseline checkpoints to Hugging Face (HF) under SeasonBench-EA.
>
> We fully understand the reviewer's concerns regarding software installation. We are now migrating the full datasets to HF under username *SauryChen/* (which may not be finished during the rebuttal period due to large dataset size). To ensure flexible downloads, the data is organized into separate repositories by NWP models and pressure levels.
>
> To facilitate community use, we will upload data download scripts to HF SeasonBench-EA, allow users to fetch selected variables or time periods directily from original climate data store website, as also discussed in our response to Q2.
>
> We have updated the Github repository to include installation instructions and package requirements. We hope these efforts can address your concerns and improve the accessibility.
> ## Additional feedbacks
> - Geopotential at surface: This variable (paramID: 129) represents the surface geopotential height, which describes the gravitational potential energy of a unit mass at a particular location on Earth's surface. Therefore, this parameter does not vary in time.
> - GraphCast is used for correction. Therefore, it corrects the outputs from numerical models rather than performing iterative forecasts.
> - Correction of other NWP models: Yes, we compare the performance using GraphCast on ECMWF and CMCC. The results are shown in Supp. Section E.3. We will update the manuscript to reference it  more clearly.
>
> We will carefully revise and correct the typos.
>
> Figure 2 shows the visual comparison between prediction and post-processing results on total precipitation, we will change to the same forecast time for a better comparison.
>
> The evaluation metrics are detailed described in Supp. Section B, we will add a clear reference in the paper.

---

> > ### Author Response · Authors · 2025-08-04
> >
> > Thank you again for the constructive suggestions and valuable comments. We hope our responses have addressed your concerns. As the discussion phase is about to close, we are much looking forward to hearing from you about any further feedback, and we will be very happy to clarify further concerns (if any).

---

> ### Author Response · Authors · 2025-08-07
>
> Dear reviewer, we have completed the upload of all datasets and model checkpoints to the Hugging Face platform under the username *SauryChen/*. Since the initial release, each individual dataset has received dozens to several hundred downloads, which highlights the relevance and potential impact of the dataset within the climate community. We hope this addresses the major concern raised in **Limitations Weaknesses Q10**.
>
> We sincerely welcome any further feedback and would be happy to clarify any remaining concerns.

---

### Official Review · Reviewer_iiqg · 2025-07-02

**Rating:** 5
**Confidence:** 4

**Summary:**

The authors construct a new benchmark dataset (called SeasonBench-EA) for seasonal-scale climate prediction in East Asia. The authors perform benchmark evaluations of (1) different time series forecast models using the reanalysis data and (2) post-processing of model ensembles. The authors discuss insights and challenges of AI model-based climate prediction.

**Dataset Code Accessibility:**

Yes

**Ethical Considerations:**

No, there are no or only very minor ethics concerns

**Final Justification:**

After reviewing the authors' rebuttal to all reviewers' comments, I believe that the authors have addressed the comments and concerns of the reviewers. I would recommend acceptance for this paper and updated my rating score from 4 to 5 accordingly.

**Limitations Weaknesses:**

1. The authors should include a table to summarize the numerical results of benchmark evaluations and indicate the best performing model in each prediction task. Currently it is difficult to draw insights and conclusions from multiple plots in Figure 3.

2. The authors should provide additional details on their process of data collection and preprocessing (e.g., identifying and removing anomalous data, imputing missing data).

3. What does it mean that “monthly climatology and persistence prediction are used as two physics baselines”? The authors should provide more context and cite the relevant literature for the general audience.

4. The multi-resolution aspect of the curated dataset hasn’t been sufficiently discussed or leveraged in the benchmark evaluations (e.g., using time series forecast models designed to handle multi-resolution data) .

5. To make the evaluations more rigorous and less prone to data bias and anomalies, the authors are recommended to perform a rolling window evaluation, where the training data spans the period [1940, X], validation data [X+1, Y] and test data [Y+1, Y+4].

**Strengths Contributions:**

1. The paper is clearly written and relatively easy to understand.

2. The curated dataset provides a useful benchmark for time series forecast models for climate prediction, a domain both technically challenging and practically relevant.

3. The authors provide helpful comparison of their curated dataset against existing ones.

4. The authors performed holistic and rigorous benchmark evaluations with a diverse set of metrics, including deterministic, probabilistic, and hindcast metrics.

---

> ### Author Rebuttal · Authors · 2025-07-31
>
> # Author response to Reviewer iiqg
> We sincerely thank Reviewer iiqg for the constructive and valuable comments. We will revise the manuscript and supplementary as follows.
> ## Q1: Lack of summary table for benchmark results
> We appreciate the reviewer's suggestion and agree that the original Figure 3, while informative, may be difficult to interpret due to the number of metrics, variables, lead times, and models involved. To improve clarity, we will add the following summary tables reporting **RMSE, ACC and absolute bias** across six lead times for five key variables. For each variable and lead time, we highlight the **best and the second-best** performing models among these approaches.
>
> For RMSE, since all models outperform the persistence baseline and underperform the climatology baseline, we report the results among the data-driven models.
>
> | Variable [RMSE $\downarrow$] | Lead time 1 | Lead time 2  | Lead time 3  | Lead time 4 | Lead time 5  | Lead time 6   |
> | -- | -- | -- | -- | -- | -- |-- |
> | tp [mm/day] | 2.00 (ViT) / 2.03 (Unet)  | 2.00 (ViT) / 2.14 (Unet) | 2.15 (ViT) / 2.25 (Unet)   | 2.25 (ViT) / 2.36 (Unet)   | 2.32 (ViT) / 2.45 (Unet)  | 2.37 (ViT) / 2.48 (Unet)   |
> | t2m [K]  | 2.28 (Unet) / 2.32 (ViT) | 2.74 (Unet) / 3.08 (ViT)  | 3.16 (Unet) / 4.06 (ViT) | 3.52 (Unet) / 5.22 (ViT)  | 3.76 (Unet) / 5.70 (FNO)  | 3.89 (Unet) / 5.84 (FNO)  |
> | t_850 [K]  | 1.95 (Unet) / 1.95 (ViT) | 2.24 (Unet) / 2.67 (ViT)   | 2.52 (Unet) / 3.63 (ViT) | 2.80 (Unet) / 4.61 (FNO)   | 3.02 (Unet) / 5.09 (FNO) | 3.17 (Unet) / 5.20 (FNO) |
> | z_500 [m²/s²]   | 362.20 (ViT) / 363.46 (Unet) | 392.71 (Unet) / 476.74 (FNO) | 431.39 (Unet) / 592.03 (FNO) | 476.75 (Unet) / 711.96 (FNO) | 505.98 (Unet) / 802.12 (FNO) | 516.84 (Unet) / 842.05 (FNO) |
> | q_700 [g/kg]  | 0.67 (ViT) / 0.71 (Unet)   | 0.75 (ViT) / 0.79 (Unet) | 0.87 (ViT) / 0.89 (Unet)  | 0.99 (Unet) / 1.05 (ViT)  | 1.05 (Unet) / 1.19 (ViT) | 1.06 (Unet) / 1.26 (ViT)  |
>
> For ACC, since climatology always yields zero, we include the persistence baseline for reference.
>
> | Variable [ACC $\uparrow$]  | Lead time 1 | Lead time 2  | Lead time 3  | Lead time 4 | Lead time 5  | Lead time 6  |
> | -- | -- | -- | -- | -- | -- |-- |
> | tp   | 0.17 (ViT) / 0.13 (Persis) | 0.12 (ViT) / 0.12 (Persis) | 0.11 (ViT) / 0.09 (Persis)  | 0.09 (ViT) / 0.06 (Persis)  | 0.07 (ViT) / 0.04 (VAE)       | 0.05 (ViT) / 0.03 (VAE)  |
> | t2m   | 0.21 (VAE) / 0.12 (Unet)   | 0.12 (VAE) / 0.00 (Unet)   | 0.07 (VAE) / -0.03 (Unet)   | 0.04 (VAE) / -0.04 (Unet)  | 0.04 (VAE) / -0.01 (Unet)     | 0.05 (VAE) / 0.02 (Unet)   |
> | t_850 | 0.18 (VAE) / 0.14 (Unet)   | 0.07 (VAE) / 0.00 (Unet)   | 0.02 (VAE) / -0.02 (Unet)   | 0.00 (VAE) / -0.04 (Unet)  | -0.01 (VAE) / -0.02 (Unet)    | 0.01 (VAE) / 0.00 (Unet)   |
> | z_500   | 0.18 (VAE) / 0.10 (FNO)  | 0.15 (VAE) / -0.03 (Unet)  | 0.09 (VAE) / -0.05 (Unet)  | 0.06 (VAE) / -0.08 (Unet)  | 0.05 (VAE) / -0.07 (Unet)   | 0.05 (VAE) / -0.03 (Unet)  |
> | q_700  | 0.25 (ViT) / 0.16 (Unet)   | 0.11 (ViT) / 0.03 (Unet)   | 0.00 (ViT) / -0.01 (Persis) | -0.06 (Unet) / -0.06 (Persis) | -0.07 (Persis) / -0.07 (Unet) | -0.05 (Unet) / -0.08 (VAE) |
>
> For absolute bias, we include both persistence and climatology to enable a more comprehensive comparison.
>
> | Variable [bias $\rightarrow$ 0] | Lead time 1| Lead time 2  | Lead time 3  | Lead time 4  | Lead time 5 | Lead time 6   |
> | -- | -- | -- | -- | -- | -- |-- |
> | tp [mm/day]  | 0.01 (FNO) / 0.04 (Clim)  | 0.05 (Clim) / 0.13 (Persis)    | 0.06 (Clim) / 0.08 (ViT)  | 0.07 (Clim) / 0.08 (ViT) | 0.08 (Clim) / 0.22 (ViT) | 0.09 (Clim) / 0.30 (Persis)  |
> | t2m [K] | 0.31 (Persis) / 0.38 (Unet)    | 0.59 (Clim) / 0.68 (Persis)  | 0.58 (Clim) / 0.96 (Unet)| 0.57 (Clim) / 1.03 (Unet)  | 0.58 (Clim) / 0.99 (Unet)   | 0.60 (Clim) / 0.85 (Unet)    |
> | t_850 [K] | 0.28 (Persis) / 0.28 (Unet)    | 0.47 (Clim) / 0.60 (Persis)    | 0.46 (Clim) / 0.81 (Unet)  | 0.45 (Clim) / 0.87 (Unet)  | 0.47 (Clim) / 0.86 (Unet) | 0.48 (Clim) / 0.76 (Unet)  |
> | z_500 [m²/s²]  | 31.89 (Persis) / 109.68 (Unet) | 73.11 (Persis) / 135.50 (Clim) | 111.04 (Persis) / 132.71 (Clim) | 132.65 (Clim) / 139.60 (Persis) | 132.96 (Clim) / 148.76 (Persis) | 134.58 (Clim) / 137.06 (Persis) |
> | q_700 [g/kg]   | 0.04 (ViT) / 0.06 (Unet)       | 0.16 (Unet) / 0.17 (Clim)  | 0.18 (Clim) / 0.26 (Persis)  | 0.19 (Clim) / 0.30 (VAE)  | 0.20 (Clim) / 0.27 (VAE)  | 0.20 (Clim) / 0.23 (VAE)  |
>
> We hope these tables make model-wise and variable-wise comparison clearer, and help readers interpret performance across prediction tasks more easily.
>
> ## Q2: Data collection and preprocessing
>
> All datasets used in SeasonBench-EA are sourced from high-quality climate data repositories, including ERA5 reanalysis (provided by ECMWF) and seasonal forecasts from numerical model ensembles (provided by C3S). These datasets have undergone rigorous quality control by their respective providers, and have been widely used in climate research. Therefore, we did not apply additional anomaly removal or missing value imputation.
>
> Our preprocessing pipeline focuses on cropping the data to the East Asia region, aligning the spatial and temporal grids between the numerical model forecasts and reanalysis data, as well as performing unit conversion and normalization. These steps are implemented in our preprocessing scripts to ensure reproducibility.
>
> For the numerical model selection, we filter models based on variable availability and temporal coverage. Some numerical models provided by C3S lack key variables, and we therefore exclude them from the benchmark. For example, the JMA model does not provide 1000 hPa data, while the UK Met Office, NCEP and BOM models do not include variables such as snow depth. As a result, we retain only the remaining numerical models that offer necessary variables. Additionally, we prioritize system versions of each numerical model that cover all calendar months and span longer temporal periods. When multiple system versions of a model are available, we select the latest version at the time of data download.
>
> We will add this information to the revised manuscript and supplementary Section A.
>
> ## Q3: Definition of physics baselines
> Following common practice in benchmarks for weather and climate prediction [1-3], climatology and persistence are two simple, non-learned baselines to provide reference performance levels. Persistence refers to using the initial condition as the forecast at all future lead times, and monthly climatology refers to the average value for each month computed over recent 30 years (i.e. 1991-2020), thereby capturing the seasonal cycle.
>
> We refer to them as physics baselines to distinguish them from data-driven models, as they rely on climatological statistics. We will clarify this definition in the revised manuscript and add the corresponding citations to Line 132.
>
> [1] Rasp, Stephan, et al. "WeatherBench: a benchmark data set for data‐driven weather forecasting." *Journal of Advances in Modeling Earth Systems* 12.11 (2020): e2020MS002203.
>
> [2] Nguyen, Tung, et al. "Climatelearn: Benchmarking machine learning for weather and climate modeling." *Advances in Neural Information Processing Systems* 36 (2023): 75009-75025.
>
> [3] Nathaniel, Juan, et al. "Chaosbench: A multi-channel, physics-based benchmark for subseasonal-to-seasonal climate prediction." *Advances in Neural Information Processing Systems* 37 (2024): 43715-43729.
>
> ## Q4: Multi-resolution aspect of the dataset
> We fully agree with this comment. While the current benchmark models do not leverage the multi-resolution nature of the dataset, we acknowledge that this is a promising direction for future research in the field of data-driven weather and climate prediction.
>
> Our dataset supports two spatial resolutions, ERA5 data is available at 0.25 deg over East Asia and 1 deg globally, while numerical model forecasts are provided at 1 degree. In our benchmark experiments, we used the 0.25 deg ERA5 data for prediction tasks to enable regional high-resolution learning, and use the 1 deg ERA5 data for post-processing tasks to match the resolution of numerical model outputs, which ensures resolution consistency within each benchmark task.
>
> As for the multi-resolution aspect, our dataset can facilitate some tasks that explicitly exploit this structure, such as:
>
> - Nested modeling: As demonstrated in [1-2], using coarse-resolution global inputs to drive fine-resolution regional models is an efficient strategy for regional weather forecasting, especially for data-driven methods. Most existing weather prediction models assume inputs and outputs to be on a fixed spatial grid. Extending these models to handle multi-resolution inputs and outputs in seasonal prediction is a valuable yet non-trivial direction, which our dataset can be well-suited to support.
> - Downscaling tasks: In particular, downscaling (or super-resolution in the AI field) precipitation and wind fields over East Asia is crucial for downstream applications such as hydrological planning and clean energy deployment. Our dataset provides the multi-resolution inputs to support such tasks.
>
> We will clarify the multi-resolution aspect of the dataset and add relevant references in the revised manuscript.
>
> [1] Oskarsson, Joel, et al. "Probabilistic weather forecasting with hierarchical graph neural networks." *Advances in Neural Information Processing Systems* 37 (2024): 41577-41648.
>
> [2] Gao, Yuan, et al. "OneForecast: a universal framework for global and regional weather forecasting." *arXiv preprint arXiv:2502.00338* (2025).
>
> ## Q5: Rolling window evaluation
> We fully agree with this comment. In response, we perform the rolling window evaluations for Unet, ViT, FNO and VAE using three additional data splits beyond the original setting. Due to the character limitation, we refer the reviewer to our response to Reviewer 5tpg *Q3: Impact of shifts in long-term historical data on model's performance* for details.

---

### Official Review · Reviewer_CQEw · 2025-07-03

**Rating:** 4
**Confidence:** 4

**Summary:**

In this paper, the authors introduce a novel SeasonBench-EA, which is a multi-source benchmark dataset for seasonal prediction, integrating ERA5 reanalysis data and ensemble forecasts. It can support both prediction and post-processing tasks. Baseline model results show limited performance, especially for precipitation, highlighting the need for new methods that incorporate physical constraints and long-range dependencies.

**Dataset Code Accessibility:**

Yes

**Ethical Considerations:**

No, there are no or only very minor ethics concerns

**Final Justification:**

Some of my concerns are addressed.

**Limitations Weaknesses:**

1. Limited baseline performance
2. Lack of multimodal inputs required for seasonal prediction
3. **Violation of double-blind policy**

**Strengths Contributions:**

1. SeasonBench-EA specifically targets 1–6 month lead-time prediction for the East Asian region
2. It integrates ERA5 reanalysis data and ensemble forecasts from multiple leading centers, providing diverse input signals and perspectives
3. Publicly available through GitHub, with clear documentation and baseline model implementations, supporting accessibility and reproducibility

---

> ### Author Rebuttal · Authors · 2025-07-31
>
> # Author response to Reviewer CQEw
>
> We thank Reviewer CQEw for the valuable comments and feedbacks on our datasets.
>
> ## Q1: Limited baseline performance
>
> The concern regarding the limited baseline performance is indeed a key challenge in the field of seasonal prediction. We explicitly acknowledge this limitation in the Conclusion section (e.g. *"our benchmark across a range of representative models reveals that the performance remains limited"*), and this observation is one of the **core motivations** for releasing this dataset and benchmark.
>
> As the submission is part of the *Datasets and Benchmarks Track*, our primary goal is not to demonstrate breakthrough model performance, but rather to establish a standard evaluation suite for the real-world seasonal prediction. To this end, we provide a diverse set of baselines, evaluated using multiple metrics and lead times, covering both prediction and post-processing tasks. These results reveal consistent performance gaps across methods and lead times, indicating that the problem remains far from solved, and is therefore well-suited for future development.
>
> We hope the reviewer appreciate that the value of our work lies not in high accuracy of baseline models, but in highlighting a performance bottleneck in this crucial yet under-explored area in the climate field, and in providing the community with the tools needed to improve upon it.
>
> ## Q2: Lack of multimodal inputs required for seasonal prediction
>
> We would like to clarify that our dataset already incorporates multimodal inputs spanning the land-ocean-atmosphere system, including 3D atmospheric variables, oceanic fields such as sea surface temperature and sea ice concentration, and land surface indicators such as soil moisture. These variables are well known to be essential for seasonal forecasting [1-3]. For example, sea surface temperature and sea ice influence large-scale teleconnection patterns, and can affect the East Asia Summer Monsoon, while snow depth and albedo over the Tibetan Plateau modulate regional moisture transport and convection. Furthermore, our dataset provides multi-source inputs, combining reanalysis data and ensemble forecasts across two resolutions, supporting future developments that can leverage this diversity.
>
> [1] Lu, Peirong, et al. "Deep learning for seasonal prediction of summer precipitation levels in eastern China." *Earth and Space Science* 10.11 (2023): e2023EA003129.
>
> [2] Tong, Xuan, and Wen Zhou. "Assessing predictive attribution in NMME forecasts of summer precipitation in eastern china using deep learning." *npj Climate and Atmospheric Science*7.1 (2024): 304.
>
> [3] Ma, Jieru, et al. "Skillful seasonal predictions of continental East-Asian summer rainfall by integrating its spatio-temporal evolution." *Nature Communications* 16.1 (2025): 273.
>
> ## Q3: Violation of double-blind policy
>
> According to the guidelines of the *NeurIPS 2025 Datasets & Benchmarks Track Call for Papers*, single-blind submissions are allowed. Our submission complies fully with the stated policy.
>
> We hope this can address the concerns raised. Please do not hesitate to contact us if there are any further questions or feedbacks regarding our work.

---

> ### Author Response · Authors · 2025-08-04
>
> Thank you for updating the score. We hope our responses have addressed your concerns. If you have any additional comments and suggestions, we would greatly appreciate your feedback and would be happy to clarify or improve further.

---

> ### Comment · Area_Chair_qk7d · 2025-08-06
>
> Dear Reviewer,
>
> Thank you for your valuable feedback. The authors have addressed your comments in their rebuttal. We kindly ask that you engage in discussion with the authors before submitting your Mandatory Acknowledgement.
>
> If your concerns have been adequately addressed in the rebuttal, please let the authors know. If your concerns remain unresolved, please communicate that clearly as well.
>
> Thank you for contributing to a fair and constructive review process at NeurIPS.

---

### Official Review · Reviewer_5tpg · 2025-07-05

**Rating:** 5
**Confidence:** 4

**Summary:**

This paper introduces SeasonBench-EA, a seasonal-scale meteorological dataset, and presents experimental results with foundational models such as ViT, VAE, and U-Net. It demonstrates the critical importance of this dataset, emphasizing that the absence of fine-grained seasonal-scale data would lead to degraded performance in long-term (seasonal) variable prediction, manifesting as prediction failures or excessively blurry outputs.

**Additional Feedback:**

Is it reasonable to train models using data spanning such an extended period (1940–2015), given that climatic characteristics in this region have undergone significant shifts over time? Could this approach compromise the model's predictive skill for future meteorological variables (i.e., the test data portion)?

**Dataset Code Accessibility:**

Yes

**Dataset Code Comments:**

The paper provides comprehensive code repositories accessible via GitHub, along with data download links, demonstrating exceptional reproducibility.

**Ethical Considerations:**

No, there are no or only very minor ethics concerns

**Final Justification:**

The author's response has addressed the issues raised in the "weaknesses" and "Additional Feedback" sections. Regarding the formatting issues of the paper, I hope the author can complete the relevant revisions in the subsequent revised version. Taking all these into account, my final rating is 5: Accept.

**Limitations Weaknesses:**

1. The format and writing of the paper lack rigor and require enhancement, such as Table 1 which exceeds the page margins. Additionally, there is an issue with the vertical spacing (vspace) between Figure 1 and the main text.
2. The notation in Equations (1) and (2) requires detailed explanation.

**Strengths Contributions:**

1. This paper presents SeasonBench-EA, a dataset of significant value to the meteorological research community. With comprehensive usage code and accessible data downloads, it effectively enhances the community's capability to analyze regional meteorological variables at seasonal mesoscales.

2. Furthermore, the study provides substantial experimental evidence demonstrating the necessity of this dataset. Complete architectural specifications of benchmark models are included in the supplementary materials, ensuring exceptional reproducibility.

---

> ### Author Rebuttal · Authors · 2025-07-31
>
> # Author Response to Reviewer 5tpg
> We sincerely thank Reviewer 5tpg for the recognition of our work and for providing constructive comments.
>
> ## Q1: Formatting and writing
> We have carefully revised the formatting issues in the manuscript. Specifically, we have resized Table 1 to ensure it fits within the page margins, and adjusted the vertical spacing around Figure 1 and other figures to improve visual consistency. We have also proofread the manuscript for clarity in writing.
>
>
> ## Q2: Notation in Equations
> Equation (1) defines the Anomaly Correlation Coefficient used in the hindcast evaluation:
>
> $A C C=\frac{\sum_{i=1}^{H\times W}\left(\Delta f-\overline{\Delta f}\right)\left(\Delta O-\overline{\Delta O}\right)}{\sqrt{\sum_{i=1}^{H\times W}\left(\Delta f-\overline{\Delta f}\right)^2 \sum_{i=1}^{H\times W}\left(\Delta O-\overline{\Delta O}\right)^2}}, \Delta f = f- \overline{f}, \Delta O = O- \overline{O}$
>
> Here, $f$ and $O$ represent the forecast and observe values at each grid point, $\overline{f}$ and $\overline{O}$ represent their climatological means over the evaluation years. The terms $\Delta f$ and $\Delta O$ are the corresponding anomalies. $H$ and $W$ indicate the number of latitude and longitude grid points, respectively.
>
> Equation (2) defines the Temporal Correlation Coefficient:
>
> $    TCC_{i,j} = \frac{\sum_{t=1}^{T} (f_t^{(i,j)} - \bar{f}^{(i,j)})(O_t^{(i,j)} - \bar{O}^{(i,j)})}{\sqrt{\sum_{t=1}^{T} (f_t^{(i,j)} - \bar{f}^{(i,j)})^2} \sqrt{\sum_{t=1}^{T} (O_t^{(i,j)} - \bar{O}^{(i,j)})^2}}$
>
> Here, $f_t^{(i,j)}$ and $O_t^{(i,j)}$ are the forecast and observe values at grid point $(i, j)$ for year $t$, while $\bar{f}^{(i,j)}$ and $\bar{O}^{(i,j)}$ are their corresponding temporal means at that grid point over the evaluation period of $T$ years.
>
> We will add these clarifications in the final version of the manuscript.
>
>
> ## Q3: Impact of shifts in long-term historical data on model's performance
> We acknowledge this concern, which also aligns with Q5 raised by Reviewer iiqg. The long-term climatic shifts can impact model's generalization. We conduct rolling window evaluations for Unet, ViT, FNO and VAE using three additional data splits beyond the original setting. Specifically, we fix the validation and test period to 4 and 5 years, repsectively. The following four configurations are used, and we will update the manuscript to include the results.
>
> - Training set: 1940-2015, validation set: 2016-2019, test set: 2020-2024 (in the original manuscript)
> - Training set: 1940-2010, validation set: 2011-2014, test set: 2015-2019
> - Training set: 1940-2005, validation set: 2006-2009, test set: 2010-2014
> - Training set: 1940-2000, validation set: 2001-2004, test set: 2005-2009
>
> | Model | Variable | Metric | Lead1       | Lead2        | Lead3        | Lead4        | Lead5         | Lead6         |
> | --- | --- | --- | --- | -- | --- | -- | --- | -- |
> | UNet  | Z500     | RMSE   | 369.68±7.30 | 416.95±19.39 | 483.46±50.00 | 560.92±77.08 | 633.63±109.68 | 690.48±142.08 |
> |       |          | ACC    | 0.05±0.05   | -0.03±0.05   | -0.03±0.05   | -0.03±0.05   | -0.03±0.07    | -0.04±0.06    |
> |       | t2m      | RMSE   | 2.40±0.15   | 3.04±0.27    | 3.83±0.56    | 4.62±0.86    | 5.30±1.16     | 5.83±1.44     |
> |       |          | ACC    | 0.08±0.04   | -0.01±0.05   | -0.03±0.04   | -0.04±0.03   | -0.04±0.03    | -0.04±0.04    |
> |       | tp       | RMSE   | 2.01±0.04   | 2.17±0.11    | 2.32±0.17    | 2.46±0.20    | 2.57±0.20     | 2.62±0.19     |
> |       |          | ACC    | 0.08±0.03   | 0.03±0.04    | 0.01±0.03    | 0.01±0.03    | 0.01±0.02     | 0.01±0.02     |
>
> | Model | Variable | Metric | Lead1        | Lead2        | Lead3        | Lead4         | Lead5         | Lead6         |
> | ----- | -------- | ------ | ------------ | ------------ | ------------ | ------------- | ------------- | ------------- |
> | ViT   | Z500     | RMSE   | 360.96±20.02 | 433.71±50.57 | 516.65±88.63 | 602.61±144.18 | 672.39±204.42 | 719.49±258.28 |
> |       |          | ACC    | 0.01±0.04    | -0.08±0.07   | -0.12±0.13   | -0.14±0.15    | -0.14±0.18    | -0.14±0.19    |
> |       | t2m      | RMSE   | 2.36±0.04    | 2.85±0.19    | 3.40±0.50    | 3.95±0.93     | 4.39±1.38     | 4.70±1.76     |
> |       |          | ACC    | 0.05±0.04    | -0.05±0.06   | -0.11±0.10   | -0.13±0.12    | -0.13±0.15    | -0.13±0.16    |
> |       | tp       | RMSE   | 1.97±0.02    | 2.08±0.04    | 2.16±0.04    | 2.25±0.05     | 2.32±0.07     | 2.35±0.10     |
> |       |          | ACC    | 0.10±0.05    | 0.06±0.05    | 0.04±0.05    | 0.02±0.05     | 0.02±0.04     | 0.01±0.03     |
>
> | Model | Variable | Metric | Lead1        | Lead2        | Lead3        | Lead4        | Lead5         | Lead6         |
> | ----- | -------- | ------ | ------------ | ------------ | ------------ | ------------ | ------------- | ------------- |
> | FNO   | Z500     | RMSE   | 429.63±18.97 | 455.84±23.93 | 515.91±61.40 | 593.86±93.33 | 666.35±106.10 | 710.47±103.36 |
> |       |          | ACC    | 0.03±0.07    | -0.04±0.07   | -0.09±0.11   | -0.11±0.13   | -0.12±0.14    | -0.13±0.15    |
> |       | t2m      | RMSE   | 3.39±0.07    | 3.65±0.19    | 4.04±0.35    | 4.51±0.49    | 4.93±0.52     | 5.14±0.49     |
> |       |          | ACC    | 0.02±0.05    | -0.06±0.05   | -0.11±0.07   | -0.13±0.08   | -0.15±0.07    | -0.15±0.07    |
> |       | tp       | RMSE   | 2.06±0.03    | 2.12±0.07    | 2.22±0.12    | 2.34±0.16    | 2.45±0.18     | 2.50±0.17     |
> |       |          | ACC    | 0.06±0.02    | 0.02±0.03    | 0.01±0.03    | -0.00±0.03   | -0.01±0.02    | -0.01±0.01    |
>
> | Model | Variable | Metric | Lead1         | Lead2         | Lead3         | Lead4         | Lead5         | Lead6         |
> | ----- | -------- | ------ | ------------- | ------------- | ------------- | ------------- | ------------- | ------------- |
> | VAE   | Z500     | RMSE   | 1349.51±22.71 | 1345.48±24.75 | 1345.51±11.99 | 1342.81±15.36 | 1347.89±19.90 | 1374.90±20.72 |
> |       |          | ACC    | 0.06±0.08     | 0.05±0.07     | 0.04±0.04     | 0.03±0.03     | 0.02±0.03     | 0.02±0.03     |
> |       | t2m      | RMSE   | 9.88±0.20     | 10.08±0.23    | 10.21±0.25    | 10.28±0.28    | 10.37±0.28    | 10.61±0.24    |
> |       |          | ACC    | 0.10±0.11     | 0.05±0.09     | 0.03±0.08     | 0.02±0.08     | 0.02±0.07     | 0.02±0.07     |
> |       | tp       | RMSE   | 2.85±0.02     | 3.10±0.07     | 3.27±0.04     | 3.36±0.04     | 3.39±0.04     | 3.41±0.03     |
> |       |          | ACC    | 0.02±0.03     | 0.00±0.02     | 0.01±0.02     | 0.01±0.02     | 0.01±0.02     | 0.01±0.01     |
>
> In addition to the experiments using long-term training data starting from 1940, we further investigate the impact of training period by training the models with more recent observations (1979-2015), while keeping the same validation and test periods. Although recent data may better reflect contemporary climatic characteristics and benefit from improved observational quality, the substantially reduced training sample size (around 1/2 of the origin) results in less stable predictions, particularly for models such as U-Net and FNO. We will also include these results in the revised manuscript to provide a more complete picture of training data choices and their corresponding results. In the following table, each value is reported as <training from 1940> / < training from 1979>. Due to character limitations, we report only RMSE for comparison.
>
> | Model | variable | Metric | Lead1         | Lead2         | Lead3         | Lead4         | Lead5         | Lead6         |
> | ----- | -------- | ------ | ------------- | ------------- | ------------- | ------------- | ------------- | ------------- |
> | UNet  | Z500     | RMSE   | 363.46/383.12 | 392.71/420.72 | 431.39/486.44 | 476.75/568.73 | 505.98/637.46 | 516.84/690.30 |
> |       | t2m      | RMSE   | 2.28/2.55     | 2.74/3.04     | 3.16/3.71     | 3.52/4.44     | 3.76/5.05     | 3.89/5.52     |
> |       | tp       | RMSE   | 2.03/2.07     | 2.14/2.21     | 2.25/2.41     | 2.36/2.61     | 2.45/2.77     | 2.48/2.88     |
>
> | Model | variable | Metric | Lead1         | Lead2         | Lead3         | Lead4         | Lead5         | Lead6          |
> | ----- | -------- | ------ | ------------- | ------------- | ------------- | ------------- | ------------- | -------------- |
> | ViT   | Z500     | RMSE   | 362.20/333.27 | 486.07/396.23 | 636.17/488.80 | 805.31/643.31 | 964.11/787.97 | 1092.61/903.55 |
> |       | t2m      | RMSE   | 2.32/2.57     | 3.08/3.28     | 4.06/4.15     | 5.22/5.38     | 6.34/6.55     | 7.23/7.38      |
> |       | tp       | RMSE   | 2.00/2.09     | 2.09/2.28     | 2.15/2.45     | 2.25/2.58     | 2.32/2.66     | 2.37/2.70      |
>
> | Model | variable | Metric | Lead1         | Lead2         | Lead3         | Lead4          | Lead5          | Lead6          |
> | ----- | ---- | -- | --- | --- | --- | -- | --- | --- |
> | FNO   | Z500     | RMSE   | 407.58/624.46 | 476.74/801.21 | 592.03/988.54 | 711.96/1115.33 | 802.12/1180.62 | 842.05/1209.36 |
> |       | t2m      | RMSE   | 3.36/5.58     | 3.90/7.00     | 4.55/8.34     | 5.23/9.27      | 5.70/9.75      | 5.84/9.95      |
> |       | tp       | RMSE   | 2.09/2.63     | 2.19/2.86     | 2.34/3.15     | 2.52/3.37      | 2.67/3.48      | 2.73/3.52      |
>
> | Model | variable | Metric | Lead1           | Lead2           | Lead3           | Lead4           | Lead5           | Lead6           |
> | ----- | -------- | ------ | --------------- | --------------- | --------------- | --------------- | --------------- | --------------- |
> | VAE   | Z500     | RMSE   | 1319.22/1339.89 | 1318.14/1322.90 | 1342.03/1352.56 | 1341.41/1364.74 | 1340.69/1367.13 | 1365.75/1381.68 |
> |       | t2m      | RMSE   | 9.67/9.81       | 10.08/10.06     | 10.31/10.46     | 10.31/10.50     | 10.33/10.55     | 10.56/10.66     |
> |       | tp       | RMSE   | 2.86/2.87       | 3.10/3.14       | 3.29/3.31       | 3.39/3.40       | 3.42/3.43       | 3.43/3.43       |

---

> > ### Comment · Reviewer_5tpg · 2025-08-07
> >
> > Thank you for the author's thoughtful response, which has addressed most of my concerns. I believe my current rating is reasonable, and thus I will maintain my rating at this time.

---

> ### Comment · Area_Chair_qk7d · 2025-08-06
>
> Dear Reviewer,
>
> Thank you for your valuable feedback. The authors have addressed your comments in their rebuttal. We kindly ask that you engage in discussion with the authors before submitting your Mandatory Acknowledgement.
>
> If your concerns have been adequately addressed in the rebuttal, please let the authors know. If your concerns remain unresolved, please communicate that clearly as well.
>
> Thank you for contributing to a fair and constructive review process at NeurIPS.

---

### Note · Authors · 2025-08-13

Dear PC, SAC, AC and Reviewers,

We sincerely thank you for the time and effort devoted to our submission, and for your detailed, constructive feedback.

We are grateful for the recognition from all reviewers. **Reviewer 5tpg** highlights that our dataset is of significant value to the meteorological reserach community, and provides substantial experimental evidence supporting its necessity. **Reviewer CQEw** acknowledges that the dataset offers diverse input signals and perspectives. **Reviewer iiqg** notes that we have conducted rigorous benchmark evaluations with a diverse set of metrics. **Reviewer fQ8y** acknowledges that our work tackles a gap in weather prediction by providing baselines that span multiple backbone architectures and evaluation metrics commonly used in deep learning for weather prediction.

During the discussion phase, **Reviewer 5tpg** maintains a score of 5, and **Reviewer iiqg** increased the score from 4 to 5.

Although we do not have the opportunity to engage directly with Reviewer fQ8y and Reviewer CQEw during the discussion, we hope our responses have addressed their concerns. Especially, we have addressed the concerns we believe were most critical to their evaluations. For **Reviewer fQ8y**, the major concern—limited dataset accessibility (Q10)—has been fully resolved by uploading the datasets, model checkpoints, and download scripts to the Hugging Face platform, in addition to the original availability on the Harvard Dataverse and Baidu Net Disk. Since release, each dataset split (offered in smaller units for easier download and selective use) has received dozens to several hundred downloads, indicating strong interest and potential impact within the climate community. For **Reviewer CQEw**, we believe the main reason for the low score was the violation of the double-blind policy (Q3). However, under this year’s updated policy, single-blind submissions are permitted.


In conclusion, we will incorporate these valuable suggestions provided by the reviewers into our revised manuscript. We would like to emphasize that, to the best of our knowledge, our work is the first benchmark tackling seasonal prediction in East Asia, integrating both regional and global datasets at multiple resolutions, having both numerical model outputs and reanalysis data. We believe this work will serve as an important resource for advancing research in seasonal climate prediction.


Sincerely,

The Authors.

---

### Decision · Program_Chairs · 2025-09-18

**Decision:**

Accept (poster)

**Comment:**

This paper receives 2xBorderline Accept and 2xAccept from four reviewers. All the reviewers acknowledge the contributions of this benchmark, including its value to meteorological research, diverse perspectives, necessity for challenging and practical climate prediction. Reviewer fQ8y's initially recommended rejection, citing concerns regarding the declaration clarity and presentation completeness. Reviewers 5tpg and iiqg also mention some concerns of format, writing, and explanation. After the rebuttal, the majority of these points have been addressed. Consequently, their final recommendations lean positive side.

According to all the comments, the AC agrees with reviewers and decides to accept it, and suggests improving the full paper further in terms of reviewers’ comments in the camera-ready version.